# Preparation and Characterization of Novel Chitosan Coatings to Reduce Changes in Quality Attributes and Physiochemical and Water Characteristics of Mongolian Cheese during Cold Storage

**DOI:** 10.3390/foods12142731

**Published:** 2023-07-18

**Authors:** Xin Gao, Yuanrong Zheng, Yu Zhong, Ran Zhou, Bo Li, Ming Ma

**Affiliations:** 1College of Food and Tourism, Shanghai Urban Construction Vocational College, Shanghai 201415, China; gaoxin@succ.edu.cn (X.G.); libo@succ.edu.cn (B.L.); mma@shou.edu.cn (M.M.); 2College of Food Science and Technology, Shanghai Ocean University, Shanghai 201306, China; 3State Key Laboratory of Dairy Biotechnology, Shanghai Engineering Research Center of Dairy Biotechnology, Dairy Research Institute, Bright Dairy & Food Co., Ltd., Shanghai 200436, China; zhengyuanrong@brightdairy.com; 4Department of Food Science and Technology, Shanghai Jiao Tong University, Shanghai 200240, China; zhongyu@sjtu.edu.cn; 5Quality Supervision, Inspection and Testing Center for Cold Storage and Refrigeration Equipment, Ministry of Agriculture, Shanghai 201306, China

**Keywords:** O-carboxymethyl chitosan, Mongolian cheese, preservation, protein, non-protein nitrogen, water mobility

## Abstract

The objective of this study was to evaluate the effect of O-carboxymethyl chitosan coating on microbiological, physiochemical, and water characteristics of Mongolian cheese during refrigerated storage. O-carboxymethyl chitosan coatings, particularly at 1.5%, improved cheese preservation by significantly inhibiting microbial growth, reducing changes in protein and non-protein nitrogen, and preserving pH and titratable acidity. For texture profile analysis (TPA), the hardness, gumminess, and chewiness in O-CMC treatments were significantly more stable than those in the control during storage. In addition, the relaxation component and image of nuclear magnetic resonance (NMR) were used to analyze the internal water mobility of the cheese during storage. Compared with other treatments, the 1.5% O-carboxymethyl chitosan coating had the best overall preserving effect during storage. O-carboxymethyl chitosan coating could be used in cheese preservation applications and could extend the shelf life of Mongolian cheese. The cheese coated with 1.5% O-carboxymethyl chitosan coating ranked the highest in acceptability at the end of the storage period.

## 1. Introduction

Mongolian cheese, also named milk tofu, is a kind of ready-to-eat cheese made from milk via acid coagulation and has high nutritional values [1,2]. However, Mongolian cheese is a high-moisture, high-protein dairy product that is easily contaminated by microorganisms during storage, similar to mozzarella cheese [3]. The most common spoilage is the growth of molds and yeasts, leading to the deterioration of the cheese and the shortening of its shelf life [4,5]. Previous studies suggested that the textural changes of cheese were related to the degradation of protein during storage [6]. For sale and storage, the packaging of cheese is necessary to prevent post-processing contamination and spoilage from affecting its safety and quality [7]. Taking into account ecological issues, some biodegradable polymers with diverse chemical properties and various structures have been gradually introduced into consumer products. They not only display biodegradability that alleviates ecological stress, but also provide more selectivity for food-preserving packaging [8]. In addition, edible films/coatings as an emerging sector in food packaging have the advantage of biodegradability and provide protection of food quality [9,10].

The use of edible packaging materials in food can potentially slow quality changes (e.g., moisture mobility) and extend shelf life [11]. Previous studies indicated that O-carboxymethyl chitosan (O-CMC) is a high-grade derivative of chitosan with water solubility [12,13], having no taste [14]. So, in this study, we selected O-CMC as the coating material. Correspondingly, O-CMC has better antimicrobial activities than chitosan because of the substitution of the C_6_-OH of chitosan with the acetyl group, which increases the protonation of the amino group in the C_2_-NH_2_ group in the presence of the new carboxyl ion, leading to an increase in NH_3_^+^ groups [15,16,17]. Previous studies indicated that O-CMC has no toxic effects on human health [18]. In addition, O-CMC has a wide pH range, indicating that it has a wider range of application as a novel antibacterial packaging material than chitosan and other chitosan derivatives [15]. In the present study, we hypothesized that an O-CMC coating would preserve Mongolian cheese.

In the present study, Mongolian cheese was subjected to a global quality analysis during refrigerated storage. The microbiological, physiochemical, textural, and water characteristics of cheese were measured to evaluate the optimum coating solution to preserve the cheese during storage.

## 2. Materials and Methods

### 2.1. Materials

Mongolian cheese was produced in the Zhenglanqi Lifeng dairy factory (Xilinhot, Inner Mongolia, China) according to previous studies [19]. Fresh cheese was transported to the laboratory by air within 24 h at 4 °C. O-CMC (food-grade powder, 80% carboxylation degree) was obtained from Zhejiang Aoxing Biotechnology Co., Ltd. (Hangzhou, Zhejiang, China). Glycerol was purchased from Zhanyun Chemical Reagent Co., Ltd. (Shanghai, China). Tween 20 was purchased from Sinopharm Chemical Reagent Co., Ltd. (Shanghai, China). All chemical reagents used in the experiments were analytical-grade.

### 2.2. Coating Preparation

To prepare the O-CMC coating solutions, 0.5% (*w*/*v*), 1.5% (*w*/*v*), or 2.5% (*w*/*v*) O-CMC solution was dissolved in distilled water. Then, 0.25% (*w*/*v*) glycerol and 0.6% (*w*/*v*) Tween 20 were added to the three O-CMC solutions. The solutions were stirred until all the solutes were completely dissolved and left for 48 h for degassing.

### 2.3. Coating Treatment

Mongolian cheese was cut into 2 × 2 × 2 cm^3^ pieces. All cheese cubes were divided into four treatments: (1) Control: untreated cheese; (2) 0.5% O-CMC: cheese cubes were dipped in 0.5% O-carboxymethyl chitosan solution for 20 s; (3) 1.5% O-CMC: cheese cubes were dipped in 1.5% O-carboxymethyl chitosan solution for 20 s; (4) 2.5% O-CMC: cheese cubes were dipped in a 2.5% O-carboxymethyl chitosan solution for 20 s. After drying for 10 min, the cheese cubes were stored at 4 °C for later quality evaluation.

### 2.4. pH, Total Soluble Solid, Density, and Viscosity Properties of the Coating Solutions

The pH value was measured using a digital pH meter with a glass electrode (PHs-3C, Shanghai Yueping Instrument Co., Ltd., Shanghai, China). The total soluble solid (TSS) content was determined using a hand-held refractometer (RHB-18, Shanghai Yongheng Optical Instrument Manufacturing Co., Ltd., Shanghai, China). The density was measured by using the gravimetric method [20], with g/mL as the unit. All analyses were performed in triplicate. The viscosity was directly tested using a rheometer (Physica MCR 301, Anton Paar GmbH, Graz, Austria). All analyses were performed in triplicate.

### 2.5. Microbiological Analysis

Total viable counts were incubated on a nutrient agar medium at 37 °C for 48 h. Molds and yeasts were incubated on potato dextrose agar (PDA) medium in duplicate at 28 °C for 72 h [21]. The counts were enumerated using the plate counting method to determine the number of colonies forming units.

### 2.6. Protein and Non-Protein Nitrogen

The crude protein content (total nitrogen × 6.38) was determined using an automatic Kjeldahl analyzer (Kjeltectm 8400, Foss, Hilerod, Denmark). The nitrogen content in fractions of pH 4.6-souble nitrogen (pH 4.6-SN) was determined using the macro-Kjeldahl method [22,23,24]. Then, the nitrogen content in fractions of 12% trichloroacetic acid-SN (12% TCA-SN) and 5% phosphotungstic acid-SN (5% PTA-SN) was tested according to the macro-Kjeldahl method. 

### 2.7. Physicochemical Analysis

The moisture content was determined by a drying method [25]. The thiobarbituric acid (TBA) content was determined according to the method described by previous studies [26]. The pH values of Mongolian cheese were determined using a digital pH meter with a glass electrode. The titratable acidity was calculated using the ratio of lactic acid (*w*/*v*) [27].

### 2.8. Color

The color was determined using a colorimeter (CR-400, Konica Minolta Holdings, Inc., Tokyo, Japan). The L^*^, a^*^, and b^*^ parameters and the total color difference (ΔE*) were then obtained using the CIE Laboratory color scale [22]. 

### 2.9. Texture Profile Analysis (TPA)

The TPA of Mongolian cheese was measured by using a texture analyzer (TA-XT plus, Stable Micro Systems, Surrey, UK) with a P/2.5 probe to obtain a time (s)–force (N) curve (5 mm/s of probe falling speed, 10 mm/s of probe return speed, 5 s of interval time, strain 50%). The hardness, cohesiveness, springiness, gumminess, and chewiness were evaluated [28]. Each analysis was performed in twelve independent replicates.

### 2.10. Nuclear Magnetic Resonance (NMR) 

Mongolian cheese was placed in the center of the 70 mm coils of a low-field magnetic resonance imaging analyzer (MesoMR23-060H-I, Shanghai Niumag Corporation, Shanghai, China) for the NMR test. The T_2_ of cheese was determined by the Carr–Purcell–Meiboom–Gill (CPMG) pulse sequence [29]. The parameters of the CPMG included the following: 18.00 μs of 90° pulse, 35.00 μs of 180° pulse, 16 scans, and 3000 echoes. A typical T_2_ curve of Mongolian cheese contained three peaks in this study, namely T_21_, T_22_, and T_23_, which were the horizontal corresponding to each peak of the curve, with ms as the unit. M_21_, M_22_, and M_23_ referred to the percentage of the corresponding positive area of the curve to the total positive area of the curve [30].

The NMR images were obtained from the low-field magnetic resonance imaging analyzer at the beginning and end of storage. A spin echo (SE) pulse sequence was used with a repetition time (TR) of 500 ms and echo time (TE) of 20 ms.

### 2.11. Statistical Analysis

The results were subjected to analysis of variance (ANOVA) using SPSS Statistics 22 (IBM Corp., Armonk, NY, USA). Statistical significance was evaluated by Duncan’s multiple range test (*p* < 0.05). Experimental data are presented as the mean ± standard deviation (SD).

## 3. Results and Discussion

### 3.1. pH Value, Total Soluble Solid, Density, and Viscosity Properties of Coating Solutions

The pH can affect the stability and functionality of a coating layer [31]. As shown in Figure 1a, the pH value of the 2.5% O-CMC solution was 0.24 higher than that of the 0.5% O-CMC solution (*p* < 0.05). In addition, the pH values of the 1.5% and 2.5% O-CMC solutions were not statistically significantly different (*p* > 0.05); the pH of the 2.5% O-CMC solution was 0.01 higher than that of the 1.5% O-CMC solution. The 0.5%, 1.5%, and 2.5% O-CMC solutions in distilled water were all alkaline. Under these conditions, the structure of the chain in the three O-CMC solutions would be bonded via ionic bonds and interactions between -COOH groups and -NH_2_ groups in the solution, with increasing charge density [32]. In addition, an increasing number of amino groups in O-CMC would be protonated to form NH_3_^+^ groups derived from the -COOH group, which would enhance the antimicrobial ability with increasing O-CMC concentration [31,33]. Clearly, the 1.5% and 2.5% O-CMC solutions have better antimicrobial properties than the 0.5% O-CMC solution.

Total soluble solid (TSS) content is a technical parameter defined as solids dissolved in a solution [31]. This study determined that the TSS content could be used to assess the solubility of water-soluble O-CMC solutions [13]. The solubility can determine the water resistance of films or coating in moisture tests [34]. As shown in Figure 1a, these data were close to a linear increase in TSS content with an increase in concentration (y = 0.5x + 1.3167, R^2^ = 0.95) [31]. This result indicated that the solubility was not significantly different among the O-CMC solutions.

Density is related to the structure of the coating formation. Figure 1a shows that the density of the 1.5% O-CMC solution was 1.06 times that of the 0.5% O-CMC solution (*p* < 0.05), and the density of the 2.5% O-CMC solution was 0.03 g/mL higher than that of the 1.5% O-CMC solution (*p* < 0.05). Higher concentrations of O-CMC solutions may contain more hydrophilic groups, which can form a more compactly structured coating layer with water [35]. However, the results for the 2.5% O-CMC solution might have occurred because the structure of the 2.5% O-CMC solution was covered, entangled, or even stretched [36]. The results also indicated that the 1.5% O-CMC solution might be the most suitable concentration.

The viscosity of solutions affects the thickness, uniformity, and adhesion of the coating layer, such that those with a viscosity less than 700 mPa·s are suitable for casting and layer forming [37]. Solutions with higher viscosity are good for water resistance; however, their adhesion can be reduced by the increase in viscosity. As shown in Figure 1b, the viscosities of the 0.5%, 1.5%, and 2.5% O-CMC coating solutions decreased slightly with an increase in the continuous shear rate. Specifically, the viscosities of the 0.5% and 1.5% O-CMC solutions were less than 700 mPa·s, and the viscosity of the 1.5% O-CMC solution was higher than that of the 0.5% O-CMC coating solution (*p* < 0.05). This suggested that some properties of the 1.5% O-CMC solution were better than those of the 0.5% O-CMC solution. Meanwhile, the viscosity of the 2.5% O-CMC solution was around 1000 mPa·s, which could lead to the weakening of its moisturizing and antibacterial properties due to the formation of the crimping and entanglement of the chain caused by intramolecular or intermolecular hydrogen bonding. These results again suggested that the 1.5% solution was suitable as a coating solution.

### 3.2. Microbiological Analysis

Table 1 shows that total viable counts in all treatments increased significantly throughout the storage period. The total viable counts in the control were significantly higher than those in O-CMC treatments (*p* < 0.05). The phenomenon was that untreated cheese easily supported the growth of microorganisms during storage. By the 28th day, the lowest total viable counts were observed in the 1.5% O-CMC treatment. This phenomenon could be explained by the -NH^3+^ of O-CMC solution interacting with negatively charged microbial cell membranes, leading to metabolic disorder of the microorganisms and cell death [15]. The 1.5% O-CMC treatment displayed better antibacterial properties than the others.

Molds could contaminate the cheese, and yeasts could contribute to the acidification of the cheese [38]. As shown in Table 1, molds and yeasts in all treatments increase gradually with increasing storage time. Molds and yeasts did not multiply significantly at the early phase of storage. The increase in molds and yeasts in O-CMC treatments was significantly lower than that in the control with increasing storage time because of the antimicrobial effect of the O-CMC coating (*p* < 0.05). Compared with the control, relatively lower levels of molds and yeasts were observed in the cheese of the 1.5% O-CMC treatment, indicating that the 1.5% O-CMC coating could inhibit effectively the growth of molds and yeasts due to an increase in antimicrobial activity during storage.

### 3.3. Protein and Non-Protein Nitrogen Analysis

Mongolian cheese is rich in protein, and changes in protein would directly influence the flavor and texture of cheese during storage. As shown in Figure 2a, the crude protein of cheese in all treatments increased gradually with increasing storage time, which was mainly due to the decomposition of casein hydrolyzed by microbial enzymes, resulting in the production of peptides and amino acids [39]. The increases in crude protein in O-CMC treatments were lower than those in the control samples during storage (*p* < 0.05), which could be attributed to the protection offered by the O-CMC coating. In particular, the increase in crude protein in the 1.5% O-CMC treatment was the lowest throughout the storage period, indicating that the 1.5% O-CMC coating provided the best protective effect for cheese quality. The biopolymer coating formed by the O-CMC solution on the cheese surface could effectively inhibit the growth of microorganisms, thus alleviating protein decomposition.

Proteolysis is a complex biochemical reaction, and changes in proteolysis have important impacts on the shelf life of cheese [40]. The proteolytic indexes were expressed as a percentage of soluble nitrogen to total nitrogen. The pH 4.6-SN represented the primary product of proteolysis in the cheese. As shown in Figure 2b, the pH 4.6-SN value of cheese in all treatments increased gradually during storage. Compared with the control, the pH 4.6-SN values in O-CMC treatments were lower (*p* < 0.05). Generally, proteins are hydrolyzed into larger peptides by proteases and peptidases produced by bacteria and molds [41]. The lowest increase in the 1.5% O-CMC treatment might have been caused by the O-CMC-mediated inhibition of microbial growth and reduction in enzyme production. 

The 12% TCA-SN contains small peptides and amino acids hydrolyzed by lactic acid proteases and peptidases [22]. As shown in Figure 2c, the 12% TCA-SN value in all treatments increased with increasing storage time due to secondary proteolysis. The highest increase in the 12% TCA-SN value in the control occurred from day 14 to day 28 because of the appearance of bacteria and fungi. Correspondingly, the increase in the 12% TCA-SN value in O-CMC treatments was lower than that in the control during storage, and the lowest value was observed in the 1.5% O-CMC treatment (*p* < 0.05). The results showed that the degree of proteolysis in the control was higher than that in the O-CMC treatments. Thus, the O-CMC coating could effectively inhibit proteolysis and extend the shelf life of the cheese, with the 1.5% O-CMC coating showing the best inhibitory effect.

In a complex proteolysis system, the 5% PTA-SN containing free amino acids can be expressed as an index for proteolysis end-products [42]. As shown in Figure 2d, similar to the 12% TCA-SN value, the 5% PTA-SN value in the control gradually increased with increasing storage time, whereas the 5% PTA-SN value in the 1.5% O-CMC treatment tended to increase (but non-significantly) at the early stage of storage. By contrast, there was a notable increase in the control from day 14 to day 28. Correspondingly, after 28 days of storage, the 5% PTA-SN value in the control was 3.54 times that of fresh cheese. Moreover, the increase in 5% PTA-SN value in O-CMC treatments was lower than that in the control throughout the storage period, with 1.5% O-CMC treatment showing the lowest 5% PTA-SN value due to the protective effect of the coating during storage. These results suggested that the 1.5% O-CMC coating could effectively inhibit the growth of microorganisms, resulting in lower levels of proteolysis.

### 3.4. Physicochemical Analyses

As shown in Table 1, there was a significant difference in the results of moisture content among all treatments during storage. By day 7, the 0.5%, 1.5%, and 2.5% O-CMC treatments showed significant decreases by 6.16%, 3.90%, and 4.80%, respectively, while moisture content in the control declined by 6.84%, which was 1.11, 1.75, and 1.43 times higher than that in the 0.5%, 1.5%, and 2.5% O-CMC treatments (*p* < 0.05), respectively. A possible explanation was that the moisture of cheese near the epidermis in the control was rapidly evaporated at the early stage of storage. After 28 days of storage, the highest moisture content was found in cheese coated with 1.5% O-CMC layers, while the lowest was in the control. The results suggested that water loss could be suppressed effectively by the formation of a thin coating on the cheese surface. The 1.5% O-CMC coating displayed better water retention than the other treatments.

TBA reacts with low-grade fatty acids, such as aldehydes (fatty oxidation products), to generate colored compounds. These compounds can be used to assess the oxidative deterioration in cheese [43]. As shown in Table 1, the TBA values for all samples increased slightly during storage. The increase in the TBA value of the control samples was significantly higher than that in the O-CMC treatment samples by day 14 (*p* < 0.05). By day 28, the TBA values in 0.5%, 1.5%, and 2.5% O-CMC treatments were 8.50%, 16.12%, and 10.46% lower than those in the control cheese, respectively (*p* < 0.05). The results demonstrated that the O-CMC coating effectively relieved lipid oxidation caused by changes in water and microbiological growth [43]. The TBA value in the 1.5% O-CMC treatment tended to be stable after 14 days of storage, indicating the most significant mitigative effect on lipid oxidation.

As shown in Table 1, the pH of cheese gradually decreased during the entire storage period. During the early storage period, pH decreased rapidly because of the microbial decomposition of lactose [44]. After 14 days of storage, the decrease in pH in all treatments tended to slow because of the decrease in the microbial lactose utilization rate. The pH in the control decreased to 5.18 by day 28, which was the lowest pH value in comparison to other treatment samples (*p* < 0.05). The highest pH was observed in the 1.5% O-CMC treatment cheese during storage. The phenomenon was likely caused by the continuous production of acidic amino acids and free fatty acids during storage [28]. The results suggested that the 1.5% O-CMC coating could effectively inhibit proteolysis (acidic amino acids) and lipid oxidation (free fatty acids).

A change in acidity is an important factor that influences the acceptability and shelf life of cheese. As shown in Table 1, the TA in all samples gradually increased during storage. The production of lactic acid was the direct reason for the increase in TA [45]. At the end of storage, the TA in the control was 1.19, 1.36, and 1.31 times that in 0.5%, 1.5%, and 2.5% O-CMC treatments, respectively. Correspondingly, the TA for the samples of the 1.5% O-CMC treatments was lower than that for other treatment samples (*p* < 0.05), indicating that the 1.5% O-CMC coating was beneficial for preserving the cheese. Clearly, the polymer coating formed on the surface of cheese by 1.5% O-CMC solution could effectively inhibit the activity of fungi (lactic acid production) to extend the shelf life of the cheese.

### 3.5. Color Analysis

Analysis of cheese color can be used to assess cheese deterioration [7]. The results for the color parameters L*, a*, b*, and ΔE* are shown in Table 1. The L* of all the cheeses showed a significant decrease throughout the storage period. Particularly, the decline in the L* value in the 1.5% O-CMC treatment was lower than that in other treatments (*p* < 0.05). The reason for the decrease in L^*^ value might be microbiological growth on the surface of the cheese resulting in a decrease in the brightness of cheese during storage [27]. 

Cheese quality was evaluated by the change in the a* value, representing red–green, and b* value, representing yellow–blue [7]. As shown in Table 1, a* gradually decreased toward the micro-green direction, and b* increased to yellow with increasing storage time. As shown in Table 1, the a* values in all O-CMC treatments showed a smaller decrease compared with that in the control. This phenomenon indicates that O-CMC coatings can play a certain role in protecting the color of Mongolian cheese during storage. The lowest a* values were observed in the 1.5% O-CMC treatment at the end of storage. The possible reason was that the 1.5% O-CMC coating could effectively inhibit fungal growth during storage. However, the change in the b^*^ value in the O-CMC treatment was higher than that in the control during storage, which indicated that the O-CMC coating has a slight effect on the yellow–blue change of the cheese surface. In addition, the ∆E* values in all samples generally increased during the storage of Mongolian cheese. The increase in ∆E* could be ascribed to the decrease in luminosity during storage [46]. 

### 3.6. TPA

Textural properties are an index of quality that influences consumer desire, taste, and acceptance of food [47]. As shown in Table 2, the hardness of cheese in all treatments exhibited a decreasing trend with increasing storage time, which correlated positively with proteolysis degree during storage. In addition, the hardness of cheese in the control was significantly lower than that in O-CMC treatments, and the highest hardness was observed in the 1.5% O-CMC treatment, which indicated that the O-CMC coating can efficiently control changes in the internal structure of cheese protein to maintain hardness. However, it is found that the cohesiveness and springiness of cheese had no significant difference between the control and O-CMC treatments (*p* > 0.05). Gumminess and chewiness had a decreasing trend during storage. In addition, the coating treatment had a better effect on the gumminess and chewiness of Mongolian cheese compared with the control samples. From the foregoing results, it can be concluded that the textural properties of Mongolian cheese were affected by the coating treatment, and the texture data of the cheese coated with 1.5% O-CMC were less changed than those of the other treatments.

### 3.7. NMR

The T_2_ indicates various water statuses and has direct relations with the water mobility in samples [48]. The transverse relaxation time (T_2_) was defined as the amount of the ratio of the multiplying result of the abscissa data and the corresponding ordinate results to the total positive area [30]. Water in cheese interacts with the protein network structure, which can be accurately expressed by NMR relaxation because cheese is a kind of heterogeneous system [49]. The various status of water generated signal intensities with different relaxation times. Inside Mongolian cheese, there were various statuses of water as follows: water weakly, but closely, bound to adipocytes and proteins (bound water) (T_21_); immobilized water retained inside the protein matrix (T_22_); free water present in the large channel of casein (T_23_) [50]. 

Water distribution and the T_2_ of Mongolian cheese are shown in Table 3. M_2_ refers to the corresponding water fraction in T_2_ [48]. The M_21_ increased slightly at the end of storage compared with that of fresh cheese due to the combination of a small amount of free water with fat globules [51]. According to the results (Table 3), immobilized water was the main water state of Mongolian cheese. By day 28, the M_22_ of the cheese in the control was lower than that of fresh cheese, and 1.5% and 2.5% O-CMC treatments had higher M_22_ which was close to that of fresh cheese. Correspondingly, the T_22_ slightly decreased during storage due to the increase in cheese hydrophobicity and shrinking of the protein matrix, which led to water dispersal in the highly hydrated casein matrix. Then, the reduction in T_22_ in O-CMC treatments was lower than that in the control, and longer T_22_ in the 1.5% O-CMC treatment was close to that of fresh cheese. This phenomenon showed that the O-CMC coating could effectively suppress the separation of water from the protein matrix. In addition, the free water fraction M_23_ increased during storage because of proteolysis, which indicated the water became more mobile during storage. A change in water retention is mainly represented by the mutual conversion and migration of free water and immobilized water. M_23_ values for samples in the 1.5% O-CMC treatment were lower than those in the control (*p* < 0.05). Then, the T_23_ in all treatments significantly decreased by day 28, while there was a non-significant difference between control and O-CMC treatments due to lower content. Clearly, immobilized water was retained inside the protein matrix, which was attributed to the protection of the O-CMC coating; the 1.5% O-CMC coatings had better effects than the others.

NMR images play an important role in investigating the spatial distribution of water molecules in a food matrix visually [52]. NMR images representing the change in water distribution of Mongolian cheese at the beginning (a) and at the end of storage (b–e) are shown in Figure 3. Lighter areas in the NMR image indicate longer relaxation times and darker areas in the NMR image indicate shorter relaxation times [53]. Cheese with higher moisture content has longer relaxation times. During storage, Mongolian cheese cracked because of water loss, and there was almost no relaxation time in the cracked area. Initially, the spatial distribution of the lighter color in cheese was fairly uniform (Figure 3a). Water loss of cheese can shorten the relaxation time during storage, which would darken the color in images. Compared with fresh cheese, the T_2_ of cheese was slightly reduced in all treatments at the end of storage time (Table 3). Correspondingly, the T_2_ in the control decreased as moisture was lost from the cheese during storage, resulting in the appearance of notable black areas inside the cheese (Figure 3b). As shown in Figure 3c,e, the T_2_ of the cheese epidermis was slightly reduced, and there were fewer darker areas compared with the control (*p* < 0.05). Correspondingly, the water distribution in the 1.5% O-CMC treatment (Figure 3d) was similar to that of fresh cheese (Figure 3a). As a result, water mobility was significantly different among the control and O-CMC treatments. The 1.5% O-CMC treatment had a positive effect on reducing water mobility during storage.

## 4. Conclusions

The results of the present study supported the hypothesis that the O-CMC coating would preserve Mongolian cheese. Compared with the 1.5% and 2.5% O-CMC coatings, the 1.5% O-CMC coating could effectively inhibit microbiological growth and alleviate change in quality attributes and protein/non-protein nitrogen of Mongolian cheese. Additionally, the most stable TPA was observed in the 1.5% O-CMC treatment. NMR was used to analyze the changes in the water characteristics of Mongolian cheese. Compared with the control, water mobility in the O-CMC treatment had a small change with the increase in storage time. In conclusion, the O-CMC coating reduced various quality changes; among the three coating treatments (0.5%, 1.5%, and 2.5%), the 1.5% O-CMC coating had the greatest impact on the quality and shelf life of Mongolian cheese.

## Figures and Tables

**Figure 1 foods-12-02731-f001:**
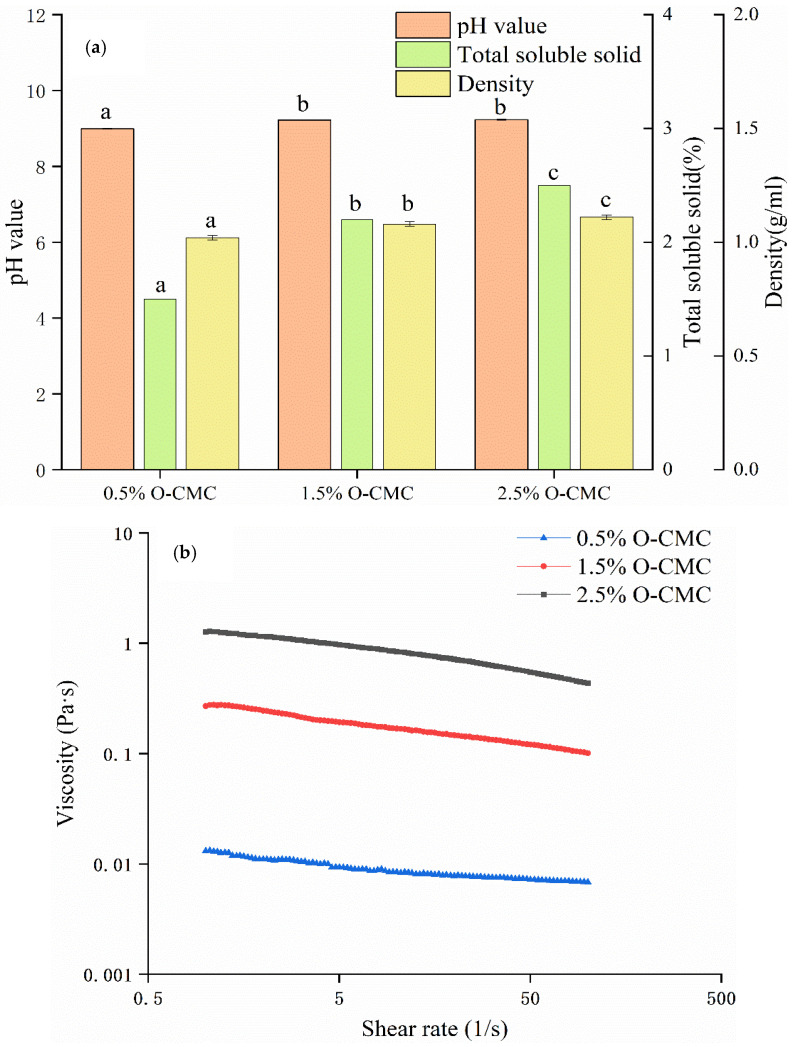
Changes in pH, total soluble solids, density (**a**), and viscosity (**b**) of 0.5%, 1.5%, and 2.5% O-carboxymethyl chitosan (O-CMC) solutions. Values with the same letter above them for pH value, total soluble solids and density are not significantly different (*p* > 0.05). Solutions: 0.5% O-CMC: 0.5 g of O-CMC dissolved in 100 mL of distilled water containing 0.25 g of glycerol and 0.6 g of Tween 20; 1.5% O-CMC: 1.5 g of O-CMC dissolved in 100 mL of distilled water containing 0.25 g of glycerol and 0.6 g of Tween 20; 2.5% O-CMC: 2.5 g of O-CMC dissolved in 100 mL of distilled water containing 0.25 g of glycerol and 0.6 g of Tween 20.

**Figure 2 foods-12-02731-f002:**
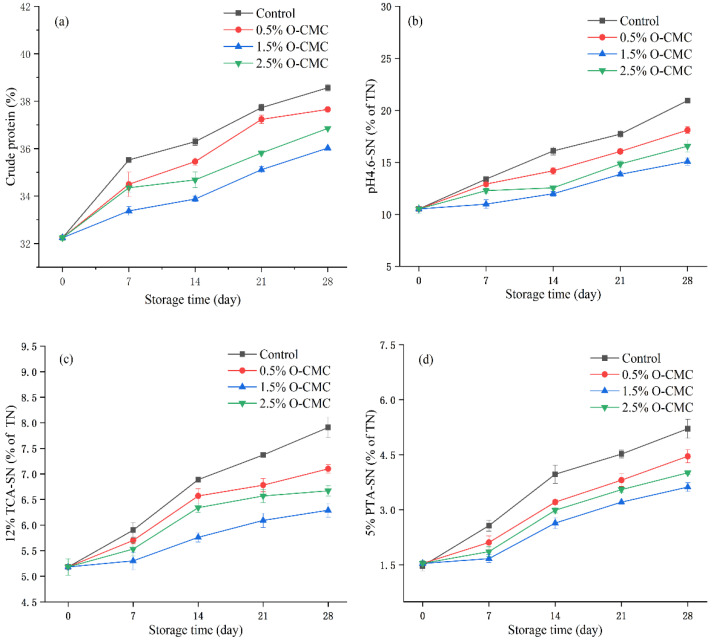
Change in crude protein (**a**), soluble nitrogen at pH 4.6 (pH 4.6-SN) (**b**), 12% trichloroacetic acid-soluble nitrogen (12% TCA-SN) (**c**), and 5% phosphotungstic acid-soluble nitrogen (5% PTA-SN) (**d**) of Mongolian cheese during storage. Control: untreated Mongolian cheese; 0.5% O-CMC: Mongolian cheese coated with 0.5% O-CMC solution; 1.5% O-CMC: Mongolian cheese coated with 1.5% O-CMC solution; 2.5% O-CMC: Mongolian cheese coated with 2.5% O-CMC solution.

**Figure 3 foods-12-02731-f003:**
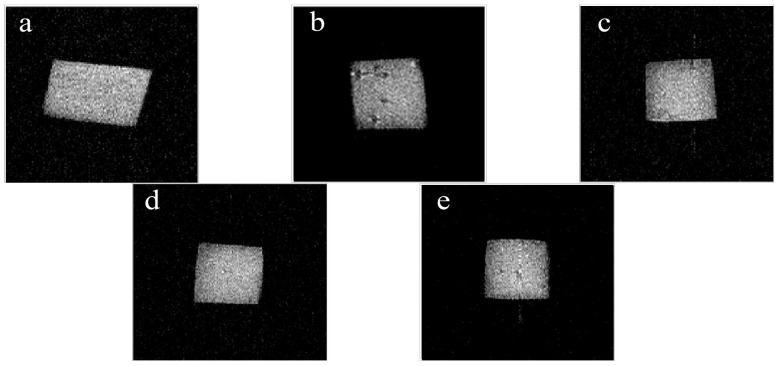
Nuclear magnetic resonance (NMR) images showing the water distribution of Mongolian cheese during storage. Untreated fresh Mongolian cheese at the beginning of storage (**a**); untreated Mongolian cheese at the end of storage (**b**); 0.5% O-CMC: Mongolian cheese coated with 0.5% O-CMC solution (**c**); 1.5% O-CMC: Mongolian cheese coated with 1.5% O-CMC solution (**d**); 2.5% O-CMC: Mongolian cheese coated with 2.5% O-CMC solution (**e**).

**Table 1 foods-12-02731-t001:** Change in total viable counts and molds and yeasts and physicochemical properties of Mongolian cheese during storage.

Parameters	Time (day)	Control	0.50% O-CMC	1.50% O-CMC	2.50% O-CMC
Total viable count (log CFU/g)	0	5.20 ± 0.14a	5.20 ± 0.14a	5.20 ± 0.14a	5.20 ± 0.14a
	7	5.45 ± 0.13a	5.39 ± 0.12a	5.27 ± 0.18a	5.28 ± 0.03a
	14	6.50 ± 0.00a	5.91 ± 0.02b	5.49 ± 0.01c	5.54 ± 0.00d
	21	6.98 ± 0.00a	6.31 ± 0.01b	5.92 ± 0.09c	6.07 ± 0.10d
	28	7.08 ± 0.01a	6.68 ± 0.07b	6.39 ± 0.01c	6.49 ± 0.01d
Molds & yeasts (log CFU/g)	0	5.80 ± 0.12a	5.80 ± 0.12a	5.80 ± 0.12a	5.80 ± 0.12a
	7	6.22 ± 0.12a	6.12 ± 0.08a	6.00 ± 0.00a	6.10 ± 0.11a
	14	6.87 ± 0.00a	6.40 ± 0.01b	6.06 ± 0.09c	6.25 ± 0.08d
	21	7.31 ± 0.00a	6.57 ± 0.05b	6.22 ± 0.06c	6.37 ± 0.09d
	28	7.35 ± 0.03a	6.84 ± 0.05b	6.43 ± 0.05c	6.59 ± 0.02d
Moisture (%)	0	51.90 ± 0.23a	51.90 ± 0.23a	51.90 ± 0.23a	51.90 ± 0.23a
	7	45.06 ± 0.18d	45.74 ± 0.13c	48.00 ± 0.21a	47.10 ± 0.17b
	14	42.69 ± 0.20d	43.60 ± 0.04c	45.83 ± 0.10a	44.65 ± 0.28b
	21	39.67 ± 0.21d	40.40 ± 0.29c	43.17 ± 0.05a	42.68 ± 0.10b
	28	36.46 ± 0.15d	37.52 ± 0.06c	40.81 ± 0.15a	39.58 ± 0.28b
TBA (mg/kg)	0	0.27 ± 0.00a	0.27 ± 0.00a	0.27 ± 0.00a	0.27 ± 0.00a
	7	0.37 ± 0.01a	0.36 ± 0.01a	0.34 ± 0.01a	0.35 ± 0.00a
	14	0.43 ± 0.01b	0.39 ± 0.00a	0.38 ± 0.01a	0.39 ± 0.01a
	21	0.46 ± 0.00c	0.43 ± 0.01b	0.38 ± 0.01a	0.42 ± 0.01b
	28	0.47 ± 0.00b	0.43 ± 0.01a	0.40 ± 0.01a	0.42 ± 0.01a
pH	0	5.47 ± 0.00a	5.47 ± 0.00a	5.47 ± 0.00a	5.47 ± 0.00a
	7	5.30 ± 0.02c	5.31 ± 0.01c	5.40 ± 0.02a	5.34 ± 0.00b
	14	5.26 ± 0.02b	5.29 ± 0.01b	5.38 ± 0.01a	5.33 ± 0.02b
	21	5.22 ± 0.01a	5.25 ± 0.01a	5.28 ± 0.02a	5.25 ± 0.01a
	28	5.15 ± 0.00b	5.18 ± 0.01b	5.22 ± 0.01a	5.20 ± 0.01a
Titratable acidity (%)	0	0.50 ± 0.06a	0.50 ± 0.06a	0.50 ± 0.06a	0.50 ± 0.06a
	7	0.88 ± 0.01a	0.80 ± 0.03a	0.77 ± 0.02b	0.77 ± 0.02b
	14	0.99 ± 0.04a	0.88 ± 0.02b	0.78 ± 0.05c	0.86 ± 0.00b
	21	1.08 ± 0.03a	0.91 ± 0.01b	0.86 ± 0.07b	0.90 ± 0.02b
	28	1.31 ± 0.08a	1.09 ± 0.07b	0.96 ± 0.04b	1.00 ± 0.04b
L^*^	0	93.11 ± 0.10a	93.11 ± 0.10a	93.11 ± 0.10a	93.11 ± 0.10a
	7	83.04 ± 0.16c	80.87 ± 0.35b	81.99 ± 1.01a	83.37 ± 1.88a
	14	82.82 ± 0.80c	80.29 ± 1.25b	84.26 ± 1.29a	82.95 ± 1.67b
	21	77.18 ± 4.09b	77.65 ± 3.92b	82.23 ± 1.12a	81.23 ± 0.68a
	28	81.01 ± 0.59b	74.97 ± 0.45c	84.31 ± 0.00a	81.21 ± 0.07b
a^*^	0	−3.71 ± 0.10a	−3.71 ± 0.10a	−3.71 ± 0.10a	−3.71 ± 0.10a
	7	−4.74 ± 0.26b	−5.04 ± 0.28a	−5.23 ± 0.15a	−5.09 ± 0.14a
	14	−5.18 ± 0.24b	−5.47 ± 0.15a	−5.39 ± 0.17a	−5.30 ± 0.16a
	21	−5.77 ± 0.10a	−5.79 ± 0.54a	−5.40 ± 0.41a	−5.77 ± 0.16a
	28	−5.79 ± 0.02d	−5.59 ± 0.13c	−5.03 ± 0.18a	−5.28 ± 0.16b
b^*^	0	13.41 ± 0.16a	13.41 ± 0.16a	13.41 ± 0.16a	13.41 ± 0.16a
	7	15.26 ± 0.20d	14.60 ± 0.38c	16.17 ± 0.10a	16.33 ± 0.46b
	14	15.94 ± 0.66b	16.73 ± 0.32b	17.74 ± 0.30a	17.04 ± 0.24a
	21	15.56 ± 0.93b	17.63 ± 0.37a	17.20 ± 0.67a	17.92 ± 0.41a
	28	15.73 ± 0.16d	18.48 ± 0.36c	18.52 ± 0.00a	19.78 ± 0.00b
∆E*	0	9.78 ± 0.14a	9.78 ± 0.14a	9.78 ± 0.14a	9.78 ± 0.14a
	7	12.74 ± 0.14a	15.4 ± 1.31b	14.51 ± 4.47b	13.5 ± 0.69ab
	14	13.82 ± 0.19a	15.78 ± 0.86b	13.97 ± 0.19a	14.29 ± 0.96ab
	21	20.66 ± 4.61a	19.8 ± 3.09a	15.88 ± 1.63a	12.45 ± 5.59a
	28	16.79 ± 3.64a	20.98 ± 0.43a	18.01 ± 3.18a	22.16 ± 3.87a

Means ± standard deviation with different lowercase letters (effect of coating) in the same row represent significant differences (*p* < 0.05). Control: untreated Mongolian cheese; 0.5% O-CMC: Mongolian cheese coated with 0.5% O-CMC solution; 1.5% O-CMC: Mongolian cheese coated with 1.5% O-CMC solution; 2.5% O-CMC: Mongolian cheese coated with 2.5% O-CMC solution.

**Table 2 foods-12-02731-t002:** Change in textural properties of Mongolian cheese during storage.

Parameters	Time (day)	Control	0.50% O-CMC	1.50% O-CMC	2.50% O-CMC
Hardness (N)	0	129.29 ± 2.11a	129.29 ± 2.11a	129.29 ± 2.11a	129.29 ± 2.11a
	7	113.14 ± 4.81d	119.04 ± 5.53c	126.51 ± 0.71a	123.32 ± 2.39b
	14	97.78 ± 3.79d	107.04 ± 1.63c	118.67 ± 1.98a	114.23 ± 4.42b
	21	96.18 ± 1.76d	106.37 ± 2.83c	113.10 ± 6.10a	83.90 ± 6.95b
	28	89.74 ± 1.66d	101.59 ± 0.86c	109.13 ± 3.69a	102.80 ± 2.43b
Cohesiveness	0	0.35 ± 0.02a	0.35 ± 0.02a	0.35 ± 0.02a	0.35 ± 0.02a
	7	0.42 ± 0.03a	0.41 ± 0.03a	0.41 ± 0.05a	0.44 ± 0.09a
	14	0.41 ± 0.04a	0.42 ± 0.02a	0.41 ± 0.03a	0.42 ± 0.04a
	21	0.42 ± 0.05a	0.37 ± 0.09a	0.38 ± 0.09a	0.41 ± 0.06a
	28	0.42 ± 0.04a	0.42 ± 0.04a	0.39 ± 0.04a	0.43 ± 0.05a
Springiness	0	0.74 ± 0.04a	0.74 ± 0.04a	0.74 ± 0.04a	0.74 ± 0.04a
	7	0.76 ± 0.04a	0.75 ± 0.03a	0.74 ± 0.02a	0.76 ± 0.03a
	14	0.74 ± 0.04a	0.72 ± 0.03a	0.74 ± 0.03a	0.75 ± 0.03a
	21	0.78 ± 0.06a	0.74 ± 0.04a	0.73 ± 0.05a	0.75 ± 0.04a
	28	0.77 ± 0.04a	0.74 ± 0.05a	0.73 ± 0.04a	0.74 ± 0.05a
Gumminess (N)	0	45.00 ± 0.05a	45.00 ± 0.05a	45.00 ± 0.05a	45.00 ± 0.05a
	7	47.28 ± 0.15d	49.39 ± 0.16c	51.92 ± 0.04b	54.37 ± 0.22a
	14	40.29 ± 0.16d	44.72 ± 0.03c	48.09 ± 0.06a	47.56 ± 0.18b
	21	40.12 ± 0.09b	39.84 ± 0.27c	42.74 ± 0.57a	34.81 ± 0.41d
	28	37.83 ± 0.06d	42.62 ± 0.04c	42.10 ± 0.16b	43.69 ± 0.11a
Chewiness (N)	0	33.45 ± 0.00a	33.45 ± 0.00a	33.45 ± 0.00a	33.45 ± 0.00a
	7	35.77 ± 0.01d	36.91 ± 0.00c	38.66 ± 0.00b	41.07 ± 0.01a
	14	29.92 ± 0.01d	32.39 ± 0.00c	35.34 ± 0.00b	35.79 ± 0.00a
	21	31.24 ± 0.01a	29.50 ± 0.01b	31.24 ± 0.03a	26.24 ± 0.02c
	28	29.15 ± 0.00d	31.55 ± 0.00b	30.67 ± 0.01c	32.43 ± 0.01a

Means ± standard deviation with different lowercase letters (effect of coating) in the same row represent significant differences (*p* < 0.05). Control: untreated Mongolian cheese; 0.5% O-CMC: Mongolian cheese coated with 0.5% O-CMC solution; 1.5% O-CMC: Mongolian cheese coated with 1.5% O-CMC solution; 2.5% O-CMC: Mongolian cheese coated with 2.5% O-CMC solution.

**Table 3 foods-12-02731-t003:** Water distribution and transverse relaxation time in Mongolian cheese at the beginning and end of storage.

	Fresh Cheese	Control	0.5% O-CMC	1.5% O-CMC	2.5% O-CMC
M_21_(%)	11.54 ± 0.06a	13.02 ± 0.69b	13.29 ± 0.09b	13.14 ± 0.18b	13.45 ± 0.73b
M_22_(%)	87.34 ± 0.85a	75.36 ± 0.38d	78.33 ± 1.56c	80.93 ± 1.19b	80.19 ± 1.42b
M_23_(%)	0.73 ± 0.21d	11.70 ± 0.41a	8.38 ± 1.64b	6.41 ± 0.46c	6.90 ± 0.27c
T_21_(ms)	0.66 ± 0.09a	0.53 ± 0.08a	0.53 ± 0.12a	0.52 ± 0.04a	0.58 ± 0.15a
T_22_(ms)	18.74 ± 0.00a	16.30 ± 0.00b	17.41 ± 0.29c	18.45 ± 0.24a	18.15 ± 3.72a
T_23_(ms)	267.34 ± 37.22a	151.99 ± 0.00e	172.69 ± 2.92c	175.25 ± 9.96b	163.48 ± 2.11d

Means ± standard deviation with different lowercase letters in the same row indicate significant differences (*p* < 0.05). Fresh cheese: untreated fresh Mongolian cheese at the beginning of storage; Control: untreated Mongolian cheese at the end of storage; 0.5% O-CMC: Mongolian cheese coated with 0.5% O-CMC solution; 1.5% O-CMC: Mongolian cheese coated with 1.5% O-CMC solution; 2.5% O-CMC: Mongolian cheese coated with 2.5% O-CMC solution. M_21_, M_22_, and M_23_ refer to the corresponding water fractions in T_21_, T_22_, and T_23_, respectively.

## Data Availability

The data presented in this study are available on request from the corresponding author.

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
