# Peer review of "Preparation and Characterization of Novel Chitosan Coatings to Reduce Changes in Quality Attributes and Physiochemical and Water Characteristics of Mongolian Cheese during Cold Storage"

_foods, 2023, doi:10.3390/foods12142731_

Round 1
Reviewer 1 Report
This paper aims to investigate the effects of novel chitosan coatings (three different concentrations of O-carboxymethyl chitosan) on the microbiological, physicochemical and water characteristics of Mongolian cheese (a fresh, very soft cheese, obtained via acid coagulation, rich of nutritive properties) during refrigerated storage. The research is interesting and well conducted, and the paper is well written. The number of tests that the authors carry out on the four experimental theses (the three coatings with different chitosan concentration and the control) are truly many and interesting, and provide an important picture of the capacity of this coat of maintaining unaltered the properties and the quality of the Mongolian cheese.
Also from an English language point of view, the paper is well written, and there are only a few sentences that could be better rephrased, and some little and minor corrections.
I have two more curiosities:
- the authors have considered the possibility that the coating can release some substances into the cheese? And, in this case, which is their level of toxicity (or legal levels admitted)? I think that a little sentence about this could be added somewhere in the introduction about this.
- the authors have considered the possibility that such substances released from the coating into the cheese can have some organoleptic repercussions? Also this fact could be considered with a sentence somewhere in the introduction.
Overall, the changes that should be made are all of a formal and non-substantial nature, so for this reason we recommend to the Editor a minor revision.
Detailed comments:
Page 1 line 33: “which rich nutrition”: I understand the meaning, but this concept should be better expressed.
Page 2 lines 46-52. A figure with the chemical formulas of chitosan and of O-carboxymethyl chitosan could be useful to better understand the difference between the two compounds.
Page 2 line 58: change “cheese was obtained” with “cheese were obtained”.
Materials and Methods: A little paragraph could be useful which briefly describe the cheesemaking technology of Mongolian cheese.
Page 2 line 86: change “ml” with “mL”. Please, make this change wherever for all the manuscript.
Page 3 line 103: The acronym TBA should be defined.
Page 3 line 134: it should be useful to specify the statistical formula, like, for example (it is only an example, it is not your statistical model), Yijk = µ + Si + Cj + εijk where: Yijk = dependent variable; µ = overall mean; Si = effect of §§§ (i = 1,…4; §§; §§; §§; §§); Cj =effect of §§ (j=1,….3; §§, §§, §§); εijk= residual error.
Page 4 line 144: change “via ionic bonds and the interactions” with “via ionic bonds and interactions”.
Page 4 line 151: change “to assess of the solubility” with “to assess the solubility”.
Tables 1 and 2: It should be better that every table stay in only one page, and not divided in two different pages.
Tables 1 and 2: It should be better, in both tables, to better understand the data, that some horizontal lines should be put after every 28th day of time, to separate the different parameters.
Page 8 lines 249-250: change “increased non-significantly” with “tended to increase (but non-significantly)”. It is better not to state peremptory comments when not supported by significant differences.
Page 8 line 252: change “that fresh cheese” with “that of fresh cheese”.
Page 9 line 300: change “all the cheese” with “all the cheeses”.
Page 9 line 303: change “that microbiological growth” with “that of a microbiological growth”.
Page 9 line 310: change “values was observed” with “values were observed”.
Page 9 line 316: change “that influences on consumer…” with “that influences consumer…”. “Influence” is a transitive verb. If you prefer to use the noun “influence”, you can alternatively write “that have influences on consumer…”.
Page 9 line 326: change “compares” with “compared”.
Page 9 lines 326-329: the sentence is not clear and should be rephrased.
Page 10 line 336: Briefly describe what T2 is.
Page 10 line 344: “the” after the point must be in capital letter T.
Page 10 line 345: Add a comma between “cheese” and “due”.
Page 10 line 348: Briefly describe what M2 is.
Page 10 line 350: Add a comma between “storage” and “due”.
Page 10 line 355: delete “had” before “increased”.
Page 10 line 359: change “than that in the control” with “than those in the control”.
Page 10 line 360: Add a comma between “28” and “while”.
Page 10 lines 361-363: the sentence is not clear and should be rephrased.
Page 11 line 385: change “In Figure 3c and e” with “In the Figures 3c and 3e”.
Page 11 line 386: change “with that in the control” with “with those in the control”.
Page 11 line 388: change “that fresh cheese” with “that of fresh cheese”.
Page 11 line 389: change “O-CMC treatments” with “O-CMC treatment”.
Figure 3: these figures are very interesting but too much small. Perhaps it should be possible separate them in different figures and show them in a bigger way.
Page 12 line 405: Add a comma between “changes” and “and 1.5%”.
Page 12 line 415: What about “Ethical Statements”?
From an English language point of view, the paper requires moderate changes. Generally is well written, but there are a few sentences that could be better rephrased, and some little and minor corrections.
Author Response
Response to Reviewer 1 Comments
Reviewer 1
This paper aims to investigate the effects of novel chitosan coatings (three different concentrations of O-carboxymethyl chitosan) on the microbiological, physicochemical and water characteristics of Mongolian cheese (a fresh, very soft cheese, obtained via acid coagulation, rich of nutritive properties) during refrigerated storage. The research is interesting and well conducted, and the paper is well written. The number of tests that the authors carry out on the four experimental theses (the three coatings with different chitosan concentration and the control) are truly many and interesting, and provide an important picture of the capacity of this coat of maintaining unaltered the properties and the quality of the Mongolian cheese.
Also from an English language point of view, the paper is well written, and there are only a few sentences that could be better rephrased, and some little and minor corrections.
1.I have two more curiosities:
- the authors have considered the possibility that the coating can release some substances into the cheese? And, in this case, which is their level of toxicity (or legal levels admitted)? I think that a little sentence about this could be added somewhere in the introduction about this.
Answer: Previous studies indicated that O-carboxymethyl chitosan (O-CMC) has no toxic effects on human health (Zhou, Zheng, Zhou, Hu, & Ma, 2020). According to the reviewer’s advice, the corresponding content was added on line 59-60 in the revised article.
Reference:
Zhou, R.; Zheng, Y.; Zhou, X.; Hu, Y.; Ma, M. Influence of Hot Water Treatment and O-Carboxymethyl Chitosan Coating on Postharvest Quality and Ripening in Hami Melons (Cucumis melo var. saccharinus) under Long-Term Simulated Transport Vibration. J. Food Biochem. 2020, 44, e13328. doi:10.1111/jfbc.13328.
2.- the authors have considered the possibility that such substances released from the coating into the cheese can have some organoleptic repercussions? Also this fact could be considered with a sentence somewhere in the introduction.
Overall, the changes that should be made are all of a formal and non-substantial nature, so for this reason we recommend to the Editor a minor revision.
Answer: Previous studies indicated that O-carboxymethyl chitosan (O-CMC) is a high-grade derivative of chitosan with water solubility, having no taste (Tang, Wu, Yang, & Lin, 2023). So in this study, we selected O-CMC as the coating material.
According to the reviewer’s advice, the corresponding content was added on line 54-56 in the revised article.
Reference:
Tang, X.; Wu, J.; Yang, J.; Lin, B. Preparation of Nano-Silver-Carboxymethyl Chitosan Composite and Its Effect on Fruit Preservation. J. Guangxi University 2023, 48, 425–432. doi:10.13624/j.cnki.issn.1001-7445.2023.0425.
3.Detailed comments:
Page 1 line 33: “which rich nutrition”: I understand the meaning, but this concept should be better expressed.
Answer: According to the reviewer’s advice, the content was changed to “which has high nutritional values (Zhang, Zheng, Zhou, & Ma, 2022).” on line 41 in the revised article.
Reference:
Zhang, X.; Zheng, Y.; Zhou, R.; Ma, M. Comprehensive Identification of Molecular Profiles Related to Sensory and Nutritional Changes in Mongolian Cheese during Storage by Untargeted Metabolomics Coupled with Quantification of Free Amino Acids. Food Chem. 2022, 386, 132740. doi:https://doi.org/10.1016/j.foodchem.2022.132740.
4.Page 2 lines 46-52. A figure with the chemical formulas of chitosan and of O-carboxymethyl chitosan could be useful to better understand the difference between the two compounds.
Answer: thank you for good advice. According to the reviewer’s advice, the figures with the chemical formulas of chitosan and O-carboxymethyl chitosan were shown as follows. The corresponding content was revised on line 59 in the revised article.
Fig. 1. Structure of chitosan (Cai et al., 2023).
Fig. 2. Structure of O-carboxymethyl chitosan (Yadav, Kaushik, Rao, Srivastava, & Vaya, 2023).
Reference:
Cai, M.; Duan, Y.; Shi, T.; Su, J.; Chen, K.; Ma, D.; Wang, F.; Qin, J.; Wei, S.; Gao, Z. Multiple Effects Achieved with a Single Agent of O-Carboxymethyl Chitosan Exhibiting Cross-Linking and Antibacterial Properties. Prog. Org. Coat. 2023, 175, 107345. doi:https://doi.org/10.1016/j.porgcoat.2022.107345.
Yadav, M.; Kaushik, B.; Rao, G.K.; Srivastava, C.M.; Vaya, D. Advances and Challenges in the Use of Chitosan and Its Derivatives in Biomedical Fields: A Review. Carbohyd. Polym. Technol. Appl. 2023, 5, 100323. doi:https://doi.org/10.1016/j.carpta.2023.100323.
5.Page 2 line 58: change “cheese was obtained” with “cheese were obtained”.
Materials and Methods: A little paragraph could be useful which briefly describe the cheesemaking technology of Mongolian cheese.
Page 2 line 86: change “ml” with “mL”. Please, make this change wherever for all the manuscript.
Answer: thank you for good advice. According to the reviewer’s advice, the content “cheese was obtained” was changed to “cheese were obtained” on line 66 in the revised article.
The detailed information about the cheese-making procedure of Mongolian cheese was as follows (Zhang, Zheng, Zhou, & Ma, 2022). Fresh milk was filtered and naturally acidified for 48 h to form milk curd at 15-20 ℃ with a relative humidity between 50% and 60%. Then, the floating fat was removed. The remnant curd was heated and stirred at 40-50 ℃ for 0.5-1 h to remove the whey completely. After the heating temperature increased to 80-95 ℃, milk coagulum was obtained and then molded in wooden moulds. Mongolian cheese was prepared after natural cooling. According to the reviewer’s advice, the corresponding contents were revised on line 70-71 in the revised article.
According to the reviewer’s advice, the content “ml” was changed to “mL” for all the manuscript (lines 94 and 160 in the revised article).
Reference:
Zhang, X.; Zheng, Y., Feng, J.; Zhou, R.; Ma, M. Integrated Metabolomics and High-throughput Sequencing to Explore the Dynamic Correlations between Flavor Related Metabolites and Bacterial Succession in the Process of Mongolian Cheese Production. Food Res. Int. 2022, 160, 111672. doi:https://doi.org/10.1016/j.foodres.2022.111672.
6.Page 3 line 103: The acronym TBA should be defined.
Page 3 line 134: it should be useful to specify the statistical formula, like, for example (it is only an example, it is not your statistical model), Yijk = µ + Si + Cj + εijk where: Yijk = dependent variable; µ = overall mean; Si = effect of §§§ (i = 1,…4; §§; §§; §§; §§); Cj =effect of §§ (j=1,….3; §§, §§, §§); εijk= residual error.
Answer: thank you for good advice. According to the reviewer’s advice, “Thiobarbituric acid (TBA) ” was added on line 109 in the revised article.
Yes, statistical formulas were also specified with the references (Mohammed Harun Babu, Shebana, Mohammed Harish, Kanimozhi, & Arun Kumar, 2022; Shaffer, 1999).
Reference:
Mohammed Harun Babu, R.; Shebana, M.; Mohammed Harish, R.; Kanimozhi, V.; Arun Kumar, K. 7 - Data Science: a Survey on the Statistical Analysis of the Latest Outbreak of the 2019 Pandemic Novel Coronavirus Disease (COVID-19) Using ANOVA. In U. Kose, D. Gupta, V. H. C. de Albuquerque & A. Khanna (Eds.), Data Science for COVID-19. 2022, 113–139. https://doi.org/10.1016/B978-0-323-90769-9.00001-3.
Shaffer, J. P. A Semi-Bayesian Study of Duncan's Bayesian Multiple Comparison Procedure. J. Stat. Plan. Infer. 1999, 82(1), 197–213. https://doi.org/10.1016/S0378-3758(99)00042-7.
7.Page 4 line 144: change “via ionic bonds and the interactions” with “via ionic bonds and interactions”.
Page 4 line 151: change “to assess of the solubility” with “to assess the solubility”.
Tables 1 and 2: It should be better that every table stay in only one page, and not divided in two different pages.
Tables 1 and 2: It should be better, in both tables, to better understand the data, that some horizontal lines should be put after every 28th day of time, to separate the different parameters.
Answer: thank you for good advice. According to the reviewer’s advice, the corresponding contents were revised as follows.
Page 4 line 144: the content “via ionic bonds and the interactions” was changed to “via ionic bonds and interactions” on line 147 in the revised article.
Page 4 line 151: the content “to assess of the solubility” was changed to “to assess the solubility” on line 153 in the revised article.
According to the reviewer’s advice, we have put every table in one page.
According to the reviewer’s advice, the horizontal lines have been put after every 28 days of time in Tables 1 and 2 to separate the different parameters.
8.Page 8 lines 249-250: change “increased non-significantly” with “tended to increase (but non-significantly)”. It is better not to state peremptory comments when not supported by significant differences.
Page 8 line 252: change “that fresh cheese” with “that of fresh cheese”.
Page 9 line 300: change “all the cheese” with “all the cheeses”.
Page 9 line 303: change “that microbiological growth” with “that of a microbiological growth”.
Page 9 line 310: change “values was observed” with “values were observed”.
Page 9 line 316: change “that influences on consumer…” with “that influences consumer…”. “Influence” is a transitive verb. If you prefer to use the noun “influence”, you can alternatively write “that have influences on consumer…”.
Page 9 line 326: change “compares” with “compared”.
Page 9 lines 326-329: the sentence is not clear and should be rephrased.
Page 10 line 336: Briefly describe what T2 is.
Page 10 line 344: “the” after the point must be in capital letter T.
Page 10 line 345: Add a comma between “cheese” and “due”.
Page 10 line 348: Briefly describe what M2 is.
Page 10 line 350: Add a comma between “storage” and “due”.
Page 10 line 355: delete “had” before “increased”.
Page 10 line 359: change “than that in the control” with “than those in the control”.
Page 10 line 360: Add a comma between “28” and “while”.
Page 10 lines 361-363: the sentence is not clear and should be rephrased.
Page 11 line 385: change “In Figure 3c and e” with “In the Figures 3c and 3e”.
Page 11 line 386: change “with that in the control” with “with those in the control”.
Page 11 line 388: change “that fresh cheese” with “that of fresh cheese”.
Page 11 line 389: change “O-CMC treatments” with “O-CMC treatment”.
Figure 3: these figures are very interesting but too much small. Perhaps it should be possible separate them in different figures and show them in a bigger way.
Page 12 line 405: Add a comma between “changes” and “and 1.5%”.
Answer: thank you for good advice. According to the reviewer’s advice, the corresponding contents were revised as follows.
Page 8 lines 249-250: the content “increased non-significantly” was changed to “tended to increase (but non-significantly)” on line 229 in the revised article.
Page 8 line 252: the content “that fresh cheese” was changed to “that of fresh cheese” on line 231 in the revised article.
Page 9 line 300: the content “all the cheese” was changed to “all the cheeses” on line 278 in the revised article.
Page 9 line 303: the content “that microbiological growth” was changed to “that of a microbiological growth” on line 281 in the revised article.
Page 9 line 310: the content “values was observed” was changed to “values were observed” on line 288 in the revised article.
Page 9 line 316: the content “that influences on consumer…” was changed to “that influences consumer…” on line 296 in the revised article.
Page 9 line 326: the word “compares” was changed to “compared” on line 305 in the revised article.
According to the reviewer’s advice, the sentence on lines 326-329 of page 9 was changed to “From the foregoing results, it can be concluded that the textural properties of Mongolian cheese were affected by coating treatment and the texture data of the cheese coated with 1.5% O-CMC were less change than the other treatments.” on lines 306-308 in the revised article.
According to the reviewer’s advice, the sentence on line 336 of page 10 was added to “The transverse relaxation times (T2) was defined as the amount of the ratio of the multiplying result of the abscissa data and the corresponding ordinate results to the total positive area (Zhou, et al., 2015).” on lines 311-313 in the revised article.
Page 10 line 344: the word “the” was changed to “The” on line 320 in the revised article.
Page 10 line 345: a comma was added between “cheese” and “due” on line 321 in the revised article.
According to the reviewer’s advice, the sentence on line 348 of page 10 was added to “M2 referred to the corresponding water fraction in T2 (Zhou, et al., 2015).” on line 319 in the revised article.
Page 10 line 350: a comma was added between “storage” and “due” on line 325 in the revised article.
Page 10 line 355: we have deleted “had” before “increased” on line 330 in the revised article.
Page 10 line 359: the content “than that in the control” was changed to “than those in the control” on line 333 in the revised article.
Page 10 line 360: a comma was added between “28” and “while” on line 334 in the revised article.
According to the reviewer’s advice, the sentence on lines 361-363 of page 10 was changed to “Clearly, immobilized water retained inside the protein matrix which was attributed to the protection of the O-CMC coating; and 1.5% O-CMC coatings had better effects than the others.” on lines 335–337 in the revised article.
Page 11 line 385: the phrase “In Figure 3c and e” was changed to “In the Figures 3c and 3e” on line 349 in the revised article.
Page 11 line 386: the content “with that in the control” was changed to “with those in the control” on line 350 in the revised article.
Page 11 line 388: the content “that fresh cheese” was changed to “that of fresh cheese” on line 352 in the revised article.
Page 11 line 389: the content “O-CMC treatments” was changed to “O-CMC treatment” on line 353 in the revised article.
According to the reviewer’s advice, Fig. 3 was redrawn as follows.
Page 12 line 405: a comma was added before “1.5%” on line 357 in the revised article.
FIGURE 3 Nuclear magnetic resonance (NMR) images showing the water distribution of Mongolian cheese during storage. Untreated fresh Mongolian cheese at the beginning of storage(a); untreated Mongolian cheese at the end of storage (b); 0.5% O-CMC: Mongolian cheese was coated with 0.5% O-CMC solution (c); 1.5% O-CMC: Mongolian cheese was coated with 1.5% O-CMC solution (d); 2.5% O-CMC: Mongolian cheese was coated with 2.5% O-CMC solution (e).
References:
Zhou, R.; Wang, X.; Hu, Y.; Zhang, G.; Yang, P.; Huang, B. Reduction in Hami Melon (Cucumis melo var. saccharinus) Softening Caused by Transport Vibration by Using Hot Water and Shellac Coating. Postharvest Biol. Tec. 2015, 110, 214–223.doi:https://doi.org/10.1016/j.postharvbio.2015.08.022.
9.Page 12 line 415: What about “Ethical Statements”?
Answer: according to Guide for Authors of Foods, ethical statements should be provided. The detailed information was as follows.
Ethical Statements:
Conflict of Interest: The authors declare that they do not have any conflict of interest.
Ethical Review: This study does not involve any human or animal testing.
Informed Consent: Written informed consent was obtained from all study participants.

Reviewer 2 Report
Dear Authors,
The paper is interesting from both a scientific and practical point of view. In my opinion, the experiments were conducted correctly and discussed correctly. Nevertheless, a few parts of the paper can be improved.
1. Line 85
“The density was calculated by the weight of 1 ml of solution, with g/ml as the unit. “
Can you provide a reference to this method? Pipettes can be calibrated in a similar way, but you still need a suitable adapter (you can find a methodology for such calibrations at Metler Tolledo materials). In my opinion, the measurement error in such a procedure will be very large. Maybe it's worth repeating the density measurements, e.g. with the pycnometric method?
2. Line 103
"... the method described by previous studies with slight modifications [17]. " What do the authors mean by slight modifications.
3. Lines 82 and 104
How exactly was the pH measured? Was the electrode directly inserted into the solution or directly into the cheese? Please extend the description of the methodology in this part.
4. Line 298
There is no discussion of the delta E parameter in the color analysis. This parameter indicates whether the color difference is perceptible by the human eyes. I propose to calculate these values and enter them in Table 1.
5.
All abbreviations used in the work should be explained. I have not found an explanation of the abbreviations TPA (Texture profile analysis) and TBA (2-thiobarbituric acid?).
6. Line 397
In the section Conclusions. There is no explanation of the mechanism why the coating with a solution of 1.5% O-CNC concentration has the best properties. What makes it better than the coating obtained from solutions containing 0.5% and 2.5% chitosan?
Author Response
Response to Reviewer 2 Comments
10.Reviewer2
Dear Authors,
The paper is interesting from both a scientific and practical point of view. In my opinion, the experiments were conducted correctly and discussed correctly. Nevertheless, a few parts of the paper can be improved.
- Line 85
“The density was calculated by the weight of 1 ml of solution, with g/ml as the unit. “
Can you provide a reference to this method? Pipettes can be calibrated in a similar way, but you still need a suitable adapter (you can find a methodology for such calibrations at Metler Tolledo materials).
In my opinion, the measurement error in such a procedure will be very large. Maybe it's worth repeating the density measurements, e.g. with the pycnometric method?
Answer: thanks for your good advice. The density was measured according to gravimetric method (Yang, & Xu, 2022). The corresponding content was revised on line 93 in the revised article.
Reference:
Yang, C.; Xu, C. Brief Introduction of On-Line Liquid Density Measurement Technology. Auto. Tech. its Appl. 2022, 41(9), 1–4. doi:10.20033/j.1003-7241.(2022)09-0001-04.
11.
2.Line 103
"... the method described by previous studies with slight modifications [17]. " What do the authors mean by slight modifications.
Answer: thank you for good advice. TBA content was determined according to the method described by previous studies (King, 1966). The phrase "with slight modification" was deleted.
References:
King, R.L. Oxidation of Milk Fat Globule Membrane Material. I. Thiobarbituric Acid Reaction as a Measure of Oxidized Flavor in Milk and Model Systems. J. Dairy Sci. 1962, 45, 1165–1171, doi:https://doi.org/10.3168/jds.S0022-0302(62)89590-3.
12.
- Lines 82 and 104
How exactly was the pH measured? Was the electrode directly inserted into the solution or directly into the cheese? Please extend the description of the methodology in this part.
Answer: thanks for good advice. In this study, the electrode was directly inserted into the coating solution. Similar method was also used by previous studies (Holme, Davidsen, Kristiansen, Smidsrod, 2008).
Reference:
Holme, H.K.; Davidsen, L.; Kristiansen, A.; Smidsrod, O. Kinetics and Mechanisms of Depolymerization of Alginate and Chitosan in Aqueous Solution. Carbohydr Polym, 2008, 73(4), 656–664. doi:org/10.1016/j.carbpol.2008.01.007.
13.
- Line 298
There is no discussion of the delta E parameter in the color analysis. This parameter indicates whether the color difference is perceptible by the human eyes. I propose to calculate these values and enter them in Table 1.
Answer: thank you for good advice. According to the reviewer’s advice, the total color difference (ΔE*) were added in Table 1. The corresponding content was also revised on line 115, 278 and 292 in the revised article.
14.
5.All abbreviations used in the work should be explained. I have not found an explanation of the abbreviations TPA (Texture profile analysis) and TBA (2-thiobarbituric acid?).
Answer: according to the reviewer’s advice, the explanations of the abbreviations of TPA and TBA are added on lines 109 and 117 in the revised article.
15.
- Line 397
In the section Conclusions. There is no explanation of the mechanism why the coating with a solution of 1.5% O-C,,,C concentration has the best properties. What makes it better than the coating obtained from solutions containing 0.5% and 2.5% chitosan?
Answer: thanks for your good advice, the corresponding contents in the conclusion were revised on lines 357 and 363 in the revised article. The revised conclusion was as follows.
The results of the present study supported the hypothesis that the O-CMC coating would preserve Mongolian cheese. Compared with 1.5% and 2.5% O-CMC coatings, 1.5% O-CMC coating could effectively inhibit microbiological growth, and alleviate change in quality attributes and protein/non-protein nitrogen of Mongolian cheese. Additionally, the most stable TPA was observed in 1.5% O-CMC treatment. Then, NMR was used to analyze the changes in the water characteristics of Mongolian cheese. Compared with control, water mobility in O-CMC treatment had a small change with the increasing storage time. In conclusion, O-CMC coating reduced various quality changes; and among the three coating treatments (0.5%, 1.5% and 2.5%), 1.5% O-CMC coating had the greatest impact on quality and shelf life of Mongolian cheese .

Reviewer 3 Report
The article presents the results of research aimed at a food safety problem, that of maintaining the quality of cheeses during storage by means of coating films.
The research carried out highlighted the impact that these coating films have on the inhibition of the growth of microorganisms and the induced changes in the level of protein and non-protein nitrogen that can affect the quality of the product.
The composition of the cheese coating film has been carefully studied and designed to maximize the effect of inhibiting the growth of microorganisms.
The scientific quality of the manuscript it rises to the scientific level of the Foods Journal. The technical quality of the manuscript is good in terms of how it was written and how the experimental results are presented. The style of expression reflects the scientific training of the authors being in accordance with the requirements of writing the article.
The Abstract is concise and contains sufficient information to highlight the content of the article and the Introduction section provides a clear statement of the problem studied in the present manuscript.
The Materials and methods section are well presented and appropriate for the purpose of research.
Results follow the guidelines described in the Author Guidelines and they are well presented and discussed.
References are relevant and current and follow the journal’s format.
The Conclusions of the article are relevant and clearly reflect the results of the study.
The authors must check the entire manuscript once again because there are still technical editing errors.
Author Response
Response to Reviewer 3 Comments
16.
Reviewer3
The article presents the results of research aimed at a food safety problem, that of maintaining the quality of cheeses during storage by means of coating films.
The research carried out highlighted the impact that these coating films have on the inhibition of the growth of microorganisms and the induced changes in the level of protein and non-protein nitrogen that can affect the quality of the product.
The composition of the cheese coating film has been carefully studied and designed to maximize the effect of inhibiting the growth of microorganisms.
The scientific quality of the manuscript it rises to the scientific level of the Foods Journal. The technical quality of the manuscript is good in terms of how it was written and how the experimental results are presented. The style of expression reflects the scientific training of the authors being in accordance with the requirements of writing the article.
The Abstract is concise and contains sufficient information to highlight the content of the article and the Introduction section provides a clear statement of the problem studied in the present manuscript.
The Materials and methods section are well presented and appropriate for the purpose of research.
Results follow the guidelines described in the Author Guidelines and they are well presented and discussed.
References are relevant and current and follow the journal’s format.
The Conclusions of the article are relevant and clearly reflect the results of the study.
The authors must check the entire manuscript once again because there are still technical editing errors.
Answer: thank you for good advice. The whole manuscript was checked again; and all the technical editing errors proposed by the reviewers were revised.
